# Renal Corin Is Essential for Normal Blood Pressure and Sodium Homeostasis

**DOI:** 10.3390/ijms231911251

**Published:** 2022-09-24

**Authors:** Tiantian Zhou, Shengnan Zhang, Chunyu Du, Kun Wang, Xiabing Gu, Shijin Sun, Xianrui Zhang, Yayan Niu, Can Wang, Meng Liu, Ningzheng Dong, Qingyu Wu

**Affiliations:** 1Hematology Center, Cyrus Tang Medical Institute, Collaborative Innovation Center of Hematology, State Key Laboratory of Radiation Medicine and Prevention, Suzhou Medical College, Soochow University, Suzhou 215123, China; 2MOH Key Laboratory of Thrombosis and Hemostasis, Jiangsu Institute of Hematology, Soochow University, Suzhou 215006, China

**Keywords:** ANP, corin, renal physiology, salt-sensitive hypertension, sodium homeostasis

## Abstract

Atrial natriuretic peptide (ANP)-mediated natriuresis is known as a cardiac endocrine function in sodium and body fluid homeostasis. Corin is a protease essential for ANP activation. Here, we studied the role of renal corin in regulating salt excretion and blood pressure. We created corin conditional knockout (cKO), in which the *Corin* gene was selectively disrupted in the kidney (kcKO) or heart (hcKO). We examined the blood pressure, urinary Na^+^ and Cl^−^ excretion, and cardiac hypertrophy in wild-type, corin global KO, kcKO, and hcKO mice fed normal- and high-salt diets. We found that on a normal-salt diet (0.3% NaCl), corin kcKO and hcKO mice had increased blood pressure, indicating that both renal and cardiac corin is necessary for normal blood pressure in mice. On a high-salt diet (4% NaCl), reduced urinary Na^+^ and Cl^−^ excretion, increased body weight, salt-exacerbated hypertension, and cardiac hypertrophy were observed in corin kcKO mice. In contrast, impaired urinary Na^+^ and Cl^−^ excretion and salt-exacerbated hypertension were not observed in corin hcKO mice. These results indicated that renal corin function is important in enhancing natriuresis upon high salt intakes and that this function cannot be compensated by the cardiac corin function in mice.

## 1. Introduction

Hypertension is the most common cardiovascular disease, afflicting ~40% of adults worldwide [1]. Dysregulated sodium and body fluid homeostasis plays a major role in hypertension [2,3,4]. Atrial natriuretic peptide (ANP) is a key hormone in the cardiac endocrine system that maintains the salt–water balance and normal blood pressure [5,6,7,8,9,10,11]. Variants of the *NPPA* gene, encoding the ANP precursor, have been associated with blood pressure levels and risk of cardiovascular diseases [12,13,14].

Corin is a trypsin-like enzyme of the type II transmembrane serine protease family [15,16,17]. It cleaves the ANP precursor, pro-ANP, converting it to mature ANP [18,19]. In mouse models, corin deficiency blocks ANP generation, resulting in salt-sensitive hypertension [20,21]. As indicated by the results of in vitro experiments, pro-B-type natriuretic peptide (pro-BNP) is another corin substrate [22,23,24]. However, studies in cultured cardiac myocytes and corin-deficient mice have shown that furin, but not corin, is primarily responsible for pro-BNP processing in vivo [25,26]. In humans, deleterious *CORIN* variants are associated with impaired natriuretic peptide processing, hypertension, pre-eclampsia, and heart disease [22,26,27,28,29,30,31,32]. These data indicate an important role of corin in the regulation of natriuretic peptide activity and cardiovascular function.

ANP is mainly synthesized in the heart [5,33]. Similarly, corin expression is most abundant in the atrium [15,34]. It is generally accepted that the ANP generated in the heart is responsible for promoting natriuresis and diuresis in the kidney via an endocrine mechanism [5,6,35]. However, pro-ANP/ANP mRNA and protein have also been detected in the kidney [36,37,38,39,40]. Similarly, renal corin mRNA and protein expressions have also been found in rodents and humans [15,24,36,41,42]. To date, the functional significance of renal corin and ANP expressions remains unclear.

Among human kidney segments, the highest level of corin expression was found in the proximal tubules, where pro-ANP/ANP and natriuretic peptide receptor A (NPR-A) expressions were colocalized [42]. Low levels of corin, ANP, and NPR-A expressions were also detected in medullary collecting ducts. In immunostaining, corin has been identified on the apical, but not basolateral, membrane of renal epithelial cells [42,43]. In renal reabsorption, proteins on the apical epithelial surface play a central role in the transportation of water and solutes [44,45,46,47]. These data suggest that corin-mediated ANP activation may occur in the kidney to regulate salt and water reabsorption via an autocrine mechanism.

In this study, we generated corin kidney conditional knockout (kcKO) mice and examined urinary salt excretion and blood pressure in corin kcKO and heart conditional KO (hcKO) mice fed normal- and high-salt diets. Our results indicated that renal corin is essential for normal blood pressure and salt excretion, particularly when salt intakes are high.

## 2. Results

### 2.1. Generation of Corin KcKO Mice

To examine renal corin function, we generated corin kcKO mice. In RT-PCR, *Corin* mRNA expression was detected in the heart and kidney in wild-type (WT) mice and in the heart, but not the kidney, in corin kcKO mice (Figure 1A). In the negative controls, *Corin* mRNA expression was not detected in the livers from WT or corin kcKO mice or in the kidney from corin KO mice [48] (Figure 1A and Appendix A). In Western blotting, corin protein was detected in the heart and kidney from WT, but not corin KO, mice, whereas in corin kcKO mice, corin protein was detected in the heart, but not the kidney (Figure 1B,C).

Corin is expressed on the cell surface, where it converts pro-ANP to ANP [19,49]. ANP may, in turn, bind to its receptor, locally stimulating intracellular cGMP production in an autocrine or paracrine manner. Unlike in pro-ANP processing, pro-BNP is primarily processed intracellularly by furin [25] and pro-BNP processing is not impaired in corin KO mice [26]. We measured pro-ANP and cGMP levels in the kidney and heart homogenates from the mice. In the kidneys from the male corin KO and kcKO mice, the pro-ANP levels were increased (242.9 ± 23.9 and 248.8 ± 21.5 pg/mg tissue, respectively, vs. 164.1 ± 12.2 pg/mg tissue in WT, *p*-values = 0.047 and 0.027, respectively, *n* = 6–8) (Figure 1D), while the cGMP levels (an indicator of natriuretic peptide activity) were decreased (0.10 ± 0.02 and 0.11 ± 0.01 pmol/mg tissue, respectively, vs. 0.17 ± 0.01 pmol/mg tissue in WT, *p*-values = 0.011 and 0.035, respectively, *n* = 7–8) (Figure 1E). In contrast, the pro-ANP and cGMP levels in the heart samples were similar among the WT, corin KO, and kcKO mice (Figure 1F,G). Similarly increased pro-ANP and decreased cGMP levels in the kidney, but not the heart, were found in the female corin kcKO mice (Appendix A). Together with the results from RT-PCR and Western blotting, these data indicated that the *Corin* gene was disrupted in the kidney, but not in the heart, in corin kcKO mice, and that there was a reduction in cGMP-stimulating activity in the kidneys of the corin kcKO mice.

### 2.2. Increased Blood Pressure in Corin KcKO Mice

We examined the effect of renal corin deficiency on blood pressure. Increased blood pressure was observed in 3-month-old male corin kcKO mice (116.0 ± 0.8 vs. 107.6 ± 0.7 mmHg in age- and sex-matched WT mice, *p* < 0.001, *n* = 10–15) (Figure 2A). The blood pressure in the corin kcKO mice remained high at 9 and 15 months of age (*p*-values < 0.001 vs. corresponding WT mice) (Figure 2A). As previously reported [48], blood pressure was high in corin KO mice. No significant differences in blood pressure were found between the corin KO and kcKO mice at 3, 9, or 15 months of age (Figure 2A). Similar results were found in 3-, 9-, and 15-month-old female mice (Appendix A). The results indicated that renal corin is essential for normal blood pressure in mice.

We next measured the serum cGMP, angiotensin II (Ang II), and aldosterone levels. The levels of cGMP and Ang II were similar between the male WT and corin kcKO mice, whereas the aldosterone levels were decreased in the corin kcKO mice (Figure 2B–D). In the corin KO mice, the serum cGMP and aldosterone levels were decreased, whereas the Ang II levels were comparable to those of the WT mice (Figure 2B–D). Similar results for serum cGMP, Ang II, and aldosterone levels were observed in the female WT, corin KO, and corin kcKO mice (Appendix A). These results indicated that hypertension in corin kcKO mice is unlikely related to systemically decreased cGMP or increased Ang II levels.

### 2.3. Renal Histology in Corin KcKO Mice

In hematoxylin and eosin (H&E)- and periodic acid-Schiff (PAS)-stained sections, glomerular morphology was indistinguishable among the WT, corin KO, and kcKO mice (Figure 3A). No apparent pathological changes such as necrosis, fibrosis, inflammation, or ischemia were observed. Glomerular diameters and red blood cell counts were comparable among the three groups (Figure 3A,B). The levels of urinary proteins and serum creatinine and urea were similar among the WT, corin KO, and kcKO mice (Figure 3C–E). These results indicated that corin deficiency, global or kidney-specific, does not cause major renal structural changes in mice under our experimental conditions.

### 2.4. Reduced Urinary Salt Excretion in Corin KcKO Mice

To distinguish the role of renal vs. cardiac corin in natriuresis, we included corin hcKO mice, in which the *Corin* gene was specifically disrupted in the heart [48] (Appendix A). Urinary Na^+^ and Cl^−^ excretions were similar among the WT, corin KO, kcKO, and hcKO mice fed a 0.3% NaCl diet (Figure 4A,B). When switched to a 4% NaCl diet, all four groups of the mice had increased urinary Na^+^ and Cl^−^ excretions. Compared with those of the WT mice, however, the urinary Na^+^ and Cl^−^ excretions were lower in the corin KO and kcKO mice (Figure 4A,B). In contrast, the urinary Na^+^ and Cl^−^ excretions of the corin hcKO mice were not significantly different from those of the WT mice (Figure 4A,B). These results indicated that the lack of renal, but not cardiac, corin impairs urinary Na^+^ and Cl^−^ excretions upon high salt intake in mice.

Consistent with the impaired urinary salt excretion when fed the 4% NaCl diet, the body weights of the male corin KO and kcKO mice were higher than those of the WT mice, whereas body weights were not significantly increased in the male corin hcKO mice (Figure 4C). Similar results of reduced urinary Na^+^ and Cl^−^ excretions and increased body weights were observed in the female corin KO and kcKO mice fed the 4% NaCl diet (Appendix A). These results indicated that when fed a 4% NaCl diet, impaired urinary salt excretion causes water retention and body weight gains in mice lacking renal, but not cardiac, corin.

### 2.5. Salt-Exacerbated Hypertension in Corin KcKO Mice

Corin KO mice develop salt-exacerbated hypertension [21]. We examined the effect of dietary salt on blood pressure in corin kcKO and hcKO mice. Increased blood pressure was found in the corin KO, kcKO, and hcKO mice fed the 0.3% NaCl diet (Figure 5). When switched to the 4% NaCl diet, blood pressure further increased in the corin KO and kcKO mice within one week (from 115.5 ± 0.6 to 121.2 ± 1.1 mmHg, *n* = 15–18; *p* = 0.041 and 115.1 ± 0.8 to 120.3 ± 1.1 mmHg, *n* = 11–16; *p* = 0.036, respectively) (Figure 5). Blood pressure remained high in these mice on the 4% NaCl diet up to 3 weeks and returned to the pre-4% NaCl^−^ diet level when back on the 0.3% NaCl diet for three weeks (Figure 5). In contrast, blood pressure did not increase in the WT and corin hcKO mice when switched from the 0.3% to the 4% NaCl diet (Figure 5). Similar salt-exacerbated hypertension was found in the female corin KO and kcKO, but not WT and corin hcKO, mice (Appendix A). In the 4% NaCl diet treated mice, the serum creatine levels were similar among the four groups, whereas the serum urea levels were higher in the corin KO and kcKO, but not hcKO, mice, compared with those of the WT mice (Appendix A). The serum Ang II levels in the 4% NaCl diet treated mice (Appendix A) and urine specific gravity in the 0.3% and 4% NaCl diet treated mice (Appendix A) were similar among the four groups. These results indicated a crucial role of renal, but not cardiac, corin in regulating sodium homeostasis and normal blood pressure upon high salt intake.

### 2.6. Water and Food Intakes and Urine Volume in Corin KcKO Mice

We measured the water and food intakes and urine volume in the WT, corin KO, kcKO, and hcKO mice. As shown in Figure 6, similar food and water intakes and urine volumes were found in among the four groups of mice fed the 0.3% NaCl diet. When switched to the 4% NaCl diet, the water intake and urine volume in all the groups increased within one week and remained high for another two weeks (Figure 6). In contrast, the food intakes were unchanged (Figure 6). There were no significant differences in water and food intakes or urine volume among these mice. Similar results were found in the female WT, corin KO, kcKO, and hcKO mice (Appendix A).

### 2.7. Cardiac Hypertrophy in Corin KcKO Mice

In echocardiography, the cardiac function, as measured by the ejection fraction (EF) and fractional shortening (FS), was similar in the male WT, corin KO, kcKO, and hcKO mice fed the 0.3% and 4% NaCl diets (Figure 7A,B). No cardiac hypertrophy, as measured by the ratios of heart weight (HW) to body weight (BW) (Figure 7C) or tibia length (TL) (Figure 7D), was detected in these mice fed the 0.3% NaCl diet. When fed the 4% NaCl diet for 3 weeks, increased HW to BW or TL ratios were observed in the corin KO and kcKO mice compared with those in the WT mice (Figure 7C,D). The ratios of HW to BW or TL appeared high in the corin hcKO mice fed the 4% NaCl diet. However, the differences between the WT and corin hcKO mice were not significant (Figure 7C,D). Similar results of normal cardiac function and cardiac hypertrophy (as indicated by the HW/BW ratio) were found in the female corin KO and kcKO mice fed the 4% NaCl diet (Appendix A). These results indicated that cardiac hypertrophy in the corin KO and kcKO mice probably was caused by salt-exacerbated hypertension, which did not result in functional declines during the experimental period.

## 3. Discussion

Since the discovery of the natriuretic activity in rat atrial extracts in the early 1980s [50], ANP has been known as a primary hormone in cardiac endocrine functioning, regulating sodium and body fluid homeostases [5,6,7,8,51]. Corin is essential for ANP activation [19,20]. Unlike soluble trypsin-like proteases, which can function at distant sites, corin has a membrane-anchoring domain that restricts its function at expression sites [15,52]. Corin antigen has been detected in human plasma [53,54,55,56], which probably represents inactive corin fragments shed from the cell surface [57]. Its distinct structural feature offers an opportunity to assess corin function in separate organs using tissue-specific KO mice.

To understand the role of renal corin, we created corin kcKO mice, in which corin mRNA and protein were undetectable in the kidney, as measured by RT-PCR and Western blotting, respectively. We found that the corin kcKO mice had increased blood pressure, indicating that renal corin is necessary for normal blood pressure. ANP has been shown to inhibit renin and aldosterone release from cultured rat renal juxtaglomerular and adrenal glomerulosa cells, respectively [58,59]. Intrarenal infusion of synthetic ANP suppresses renin secretion in dogs [60]. In our study, the serum Ang II levels were comparable in the WT and corin KO and kcKO mice, whereas the serum aldosterone levels were decreased in the corin KO and kcKO mice. These results are consistent with findings from another strain of corin KO mice, which had similar renin and decreased aldosterone levels in plasma compared with those in the WT mice [21]. It is unlikely, therefore, that the hypertension in corin kcKO mice was due to an overall enhanced activity of the renin–angiotensin–aldosterone system.

Natriuretic peptides evolved as a salt-lowering mechanism in primitive vertebrates living in salty waters [61,62,63]. ANP and corin homologs are conserved in all vertebrate species. Previously, impaired urinary salt excretion was found in corin KO mice fed a high-salt diet [21]. In this study, we found similarly reduced urinary Na^+^ and Cl^−^ excretions in the corin KO and kcKO mice fed a 4% NaCl diet. Apparently, the salt-lowering mechanism mediated by corin is preserved in terrestrial mammals to regulate sodium homeostasis when salt intakes fluctuate. The reason for hypertension in the corin kcKO mice fed the 0.3% NaCl diet is unclear. It may reflect dysregulated sodium homeostasis, which was magnified by the 4% NaCl diet and become detectable under our experimental conditions. Alternatively, renal corin may also contribute to other mechanisms in regulating blood pressure, which are yet to be understood.

Like the corin kcKO mice, the corin hcKO mice fed a normal-salt diet also had increased blood pressure, possibly due to impaired endocrine hypotensive and hypovolemic actions of ANP in the vessel, as previously indicated [64,65]. Surprisingly, we did not observe reduced urinary Na^+^ and Cl^−^ excretions in the corin hcKO mice fed the 4% NaCl diet. These results indicated that an autocrine or paracrine mechanism locally mediated by corin in the kidney is critical for promoting natriuresis upon high salt intakes and that this function cannot be substituted by cardiac corin function in mice.

ANP enhances natriuresis and diuresis by inhibiting multiple targets in renal segments [7], e.g., the Na^+^-K^+^-ATPase, Na^+^/H^+^ exchanger, and type IIa Na/Pi cotransporter in the proximal tubule [66,67,68], and the epithelial sodium channel and the cyclic nucleotide-gated cation channels in the medullary collecting duct [36,69]. In human and rodent kidneys, corin, ANP, NPR-A are coexpressed in the proximal tubule and the medullary collecting duct [36,42]. Further studies will be needed to determine if the renal corin function in promoting natriuretic actions along the renal segments is mediated by locally generated ANP and/or other unknown substrates in response to high dietary salt intakes.

Recently, corin, pro-ANP/ANP, and NPR-A expressions were identified in the skin eccrine sweat glands in mice and humans [48]. In corin KO mice, sweat production and Na^+^ and Cl^−^ excretion in the eccrine glands were reduced, whereas such a defect was not observed in corin hcKO mice [48]. Together with the findings of this study, the results indicated that salt excretion, either via the skin eccrine sweat glands or kidney, is primarily regulated by local corin function. In support of this idea, increased renal corin and ANP expressions were reported in rats fed a high-salt diet, probably reflecting a compensatory response to enhance salt excretion [70]. A recent longitudinal study in humans also identified *CORIN* variants associated with dietary salt sensitivity, long-term blood pressure changes, and hypertension susceptibility [71]. Conversely, reduced renal corin expression has been reported in rat kidney disease models [36,72] and in patients with chronic kidney disease and sodium retention [41]. In previous studies in animal models and patients, altered natriuretic peptide expression and processing have been found as mechanisms underlying hypertension and cardiorenal disease [73,74,75,76,77]. In a recent study, altered corin expression in the heart was also found in patients with chronic kidney disease [78]. These results are consistent with our findings, suggesting that pathological conditions impairing renal corin function may contribute to sodium retention and hypertension. Together, these data highlight a link between renal and cardiovascular diseases. More investigations are important to define the role of corin in renal physiology and disease in humans.

ANP is known to have antihypertrophic and anti-inflammatory functions in the heart, independent of its systemic blood pressure lowering actions [79,80]. Consistently, corin deficiency causes cardiac hypertrophy in mice, especially at old age [20,81]. In this study, we did not detect cardiac hypertrophy in the WT, corin KO, kcKO, or hcKO mice fed the 0.3% NaCl diet. When the mice were switched to the 4% NaCl diet for three weeks, cardiac hypertrophy was observed in the corin KO and kcKO, but not WT and hcKO, mice. These results indicated that cardiac hypertrophy in corin KO and kcKO mice is probably caused by salt-exacerbated hypertension.

There are some limitations in this study: First, we used a high-salt (4% NaCl) diet to test salt sensitivity in mice. Although 4–8% NaCl diets are commonly used as high-salt diets in rodent studies, 4% NaCl diet is ~10 fold higher than regular-salt diets in humans and mice. Additionally, we did not use Cre mice as a control. Our findings need to be verified under more physiologically relevant conditions and in different models. Second, the ELISA assay in our study is expected to detect both uncleaved pro-ANP and cleaved N-terminal pro-ANP fragments. A more detailed biochemical analysis is required to verify natriuretic peptide processing in the kidney from corin kcKO mice and to examine if there are additional corin substrates in the kidney. Finally, genetic and molecular studies in humans are critical to assess the clinical relevance of our findings in this study.

## 4. Materials and Methods

### 4.1. Mouse Models

The mouse study was approved by the Animal Use and Care Committee at Soochow University (201602A144). *Cor^flox^* mice with two loxP loci flanking exon 4 of the *Corin* gene were previously generated [48] (Appendix A). Corin KO and hcKO mice were bred, respectively, by crossing *Cor^flox^* mice with *CMV-Cre* mice (B6.C-Tg(CMV-cre)1 Cgn/J) expressing *Cre* ubiquitously [82] and mice expressing tamoxifen-inducible *Cre* driven by the heart-specific *Myh6* promoter (B6.FVB(129)-Tg(Myh6-cre/Esr^*^)1 Jmk/J) [83], as previously described [48]. To breed corin kcKO mice, *Cor^flox^* mice were bred with *Ggt1-Cre* mice (Tg(Ggt1-cre)M3Egn/J, Jackson Laboratory, Bar Harbor, ME, USA), expressing renal *Cre* under the *Ggt1* promoter [84,85] (Appendix A). The mice were crossed (>10 generations) into the C57BL/6J background. WT C57BL/6J mice were used as controls. Age- and sex-matched mice (12–16 weeks old, males and females) were used in test groups. At the end of the experiments, the mice were euthanized, and blood and tissue samples were isolated for further analyses.

### 4.2. PCR and RT-PCR

Mouse genotyping was performed with PCR using biopsy samples lysed in 100 mM Tris-HCl, 5 mM EDTA, 0.2% SDS, 200 mM NaCl, and 0.1 mg/mL proteinase K (P4850, Sigma-Aldrich, St. Louis, MO, USA). To examine *Corin* mRNA expression, total RNA isolated from tissues using an RNeasy Kit (74104, Qiagen, Dusseldorf, Germany) or a High Pure RNA Isolation Kit (R6812-02, Omega Bio-Tek, Norcross, GA, USA) was used to create cDNA with a RevertAid First Strand cDNA Synthesis Kit (K1622, Thermo Fisher Scientific, Waltham, MA, USA) for RT-PCR. Oligonucleotide primer sequences are shown in Appendix A.

### 4.3. Western Blotting

Mouse tissues were homogenized in 1% (*v*/*v*) Triton X-100, 10% (*v*/*v*) glycerin, 50 mM Tris-HCl, pH 8.0, 150 mM NaCl, and a protease inhibitor mixture (04693116001, 1:100 dilution, cOmplete, Roche Life Science, Basel, Switzerland). Proteins were analyzed by SDS-PAGE and Western blotting using a corin antibody (ab255812, 1:1000 dilution, Abcam, Cambridge, U.K.) and secondary horseradish-peroxidase-conjugated antibody (BS13278, Bioworld Technology, Nanjing, China). Western blots were reprobed using Gapdh antibody (MB001H, Bioworld Technology, Nanjing, China) as a control. Protein bands were revealed using chemiluminescent reagents (P10300, NCM Biotech, Suzhou, China) and an image analyzer (Amersham Imager 600, GE Healthcare, Chicago, IL, USA) and quantified using ImageJ-Pro Plus software (NIH).

### 4.4. Pro-ANP, cGMP, Angiotensin, and Aldosterone Measurements

Pro-ANP levels in tissue homogenates were measured by ELISA (SEA484Mu, Cloud-Clone Corp, Wuhan, China) with an antibody against pro-ANP fragments (Asn25-Arg122). Serum and tissue cGMP levels were measured by ELISA (ADI-900-013, Enzo Life Sciences, Farmingdale, NY, USA). Serum Ang II and aldosterone levels were measured by an enzyme immunoassay (EIAM-ANGII-1, RayBiotech, Norcross, GA, USA) and ELISA (E-EL-0070, Elabscience, Wuhan, China), respectively, following the manufacturers’ instructions.

### 4.5. Urinary Proteins and Electrolytes

Urine samples were collected from mice in metabolic cages. Urinary protein levels were measured using a Bradford assay (C035-2, Nanjing Jiancheng Bioengineering Institute, Nanjing, China). Urinary Na^+^ and Cl^−^ levels in mice fed normal-salt (0.3% NaCl) or high-salt (4% NaCl) diets (Suzhou Shuangshi Laboratory Animal Feed Science, Suzhou, China) were examined using an ion analyzer (6230M, Jenco Instruments, San Diego, CA, USA).

### 4.6. Serum Creatinine and Urea

Serum creatinine and urea levels were measured using an automated chemistry analyzer (BS-420, Shenzhen Mindray Bio-Medical Electronics, Shenzhen, China).

### 4.7. Renal Histology

Mouse kidneys were fixed with 4% (*v*/*v*) formalin and embedded in paraffin. Sections (4 μm thick) were prepared on adhesion microscope slides (188105, CITOTEST Scientific, Haimen, China), stained with H&E or PAS, and examined under a light microscope (DM2000 LED, Leica Geosystems, Heerbrugg, Switzerland). Glomerular red blood cells were counted to assess renal ischemia. At least five randomly selected fields in three sections from each mouse were examined, and each group included at least three mice.

### 4.8. Blood Pressure

Mouse blood pressure was measured using a computerized photoelectric tail-cuff apparatus (BP-2000, Visitech Systems, Apex, NC, USA), as previously described [86]. Briefly, the apparatus included a warmed platform with a cover to provide a calm environment. Mouse tails were placed in a cuff for blood pressure measurements. After three consecutive days of acclimation practice (<30 min for each practice per day), systolic blood pressure was measured. Mean values from 20 measuring cycles with 5 s between two cycles were recorded.

### 4.9. Water and Food Intakes and Urine Volume

Mice were singly placed in metabolic cages (Fengshi Laboratory Animal Equipment, Suzhou, China) with free access to water and food. After 5 days of acclimatization, mice were fed a 0.3% NaCl diet for 3 days, and then a 4% NaCl diet for various periods. Body weight, food consumption, and water intake were measured daily. Urine samples were collected in a container, and 24 h volume was recorded.

### 4.10. Echocardiography and Heart Weights

Mice were anesthetized with 1.5% (*v*/*v*) isoflurane in oxygen (flow rate of 0.3 L/min). Echocardiography (Vevo 2100, VisualSonics, Toronto, ON, Canada) was conducted using a 30 MHz transducer. Diastolic and systolic wall thickness, and end-diastolic and systolic dimensions of left ventricles were measured. EF and FS values were calculated. Ratios of HW to BW or TL in 9-month-old mice were calculated.

### 4.11. Statistical Analysis

Data were analyzed using Prism 8.0 (GraphPad, San Diego, CA, USA) software. If data passed the normality and equal variance tests (Anderson–Darling, D’Agostino–Pearson, Kolmogorov–Smirnov, and Shapiro–Wilk tests), one-way ANOVA followed by Tukey’s post hoc analysis was used. Otherwise, the Kruskal–Wallis test was used. Data are presented as mean ± SEM. *P*-values of < 0.05 were considered significant.

## 5. Conclusions

Electrolyte and body fluid homeostasis is critical for normal blood pressure. Here, we found that renal corin is required for normal blood pressure in mice. Disrupting the *Corin* gene in the kidney, but not in the heart, reduced urinary Na^+^ and Cl^−^ excretions and exacerbated hypertension in mice fed a high-salt diet. These results suggested that renal corin function is essential for the maintenance of salt–water balance and normal blood pressure, especially when salt intake is high. Our findings provide new insights into the role of corin in sodium homeostasis, blood pressure control, and salt-sensitive hypertension.

## Figures and Tables

**Figure 1 ijms-23-11251-f001:**
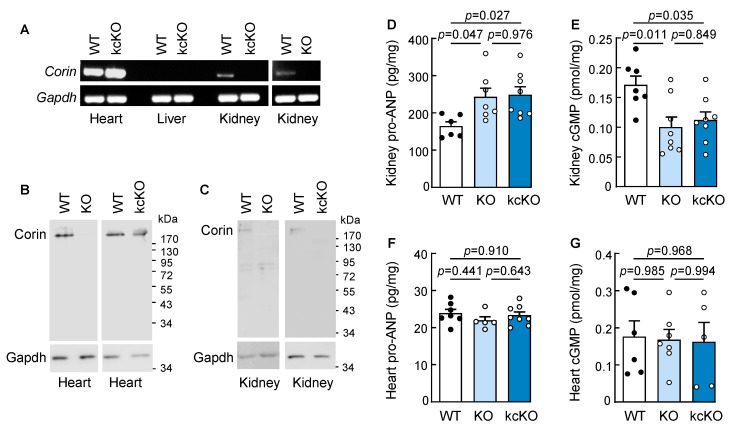
Analysis of corin kcKO mice. (**A**) RT-PCR analysis of *Corin* mRNA expression in the heart, liver (negative control), and kidney from WT, corin kcKO, and KO mice. *Gapdh* mRNA expression was used as a control. Data are representative of at least three experiments. (**B**,**C**) Western blotting of corin protein in the heart (**B**) and kidney (**C**) in WT, corin KO, and kcKO mice. Gapdh protein expression was the control. Data are representative of at least three experiments. (**D**–**G**) Levels of pro-ANP (**D**,**F**) and cGMP (**E**,**G**) in the kidney (**D**,**E**) and heart (**F**,**G**) homogenates from male WT (black dots), corin KO, and kcKO (white dots) mice were measured by ELISA (*n* = 5–8 per group). Data shown are mean ± SEM. *P*-values were analyzed by one-way ANOVA.

**Figure 2 ijms-23-11251-f002:**
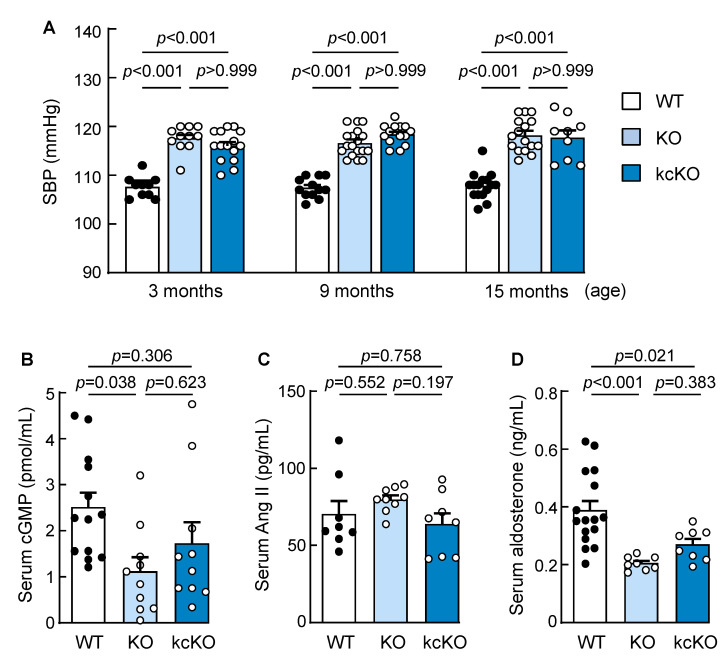
Hypertension in corin kcKO mice. (**A**) Systolic blood pressure (SBP) in male WT (black dots), corin KO, and kcKO (white dots) mice at 3, 9, and 15 months of age (*n* = 9–18). (**B**–**D**) Levels of serum cGMP (**B**), angiotensin II (Ang II) (**C**), and aldosterone (**D**) from male WT (black dots), corin KO, and kcKO (white dots) mice were measured by ELISA (**B**,**D**) or enzyme immunoassay (**C**). *n* = 8–16 per group. Data presented are mean ± SEM. *P*-values were analyzed by one-way ANOVA.

**Figure 3 ijms-23-11251-f003:**
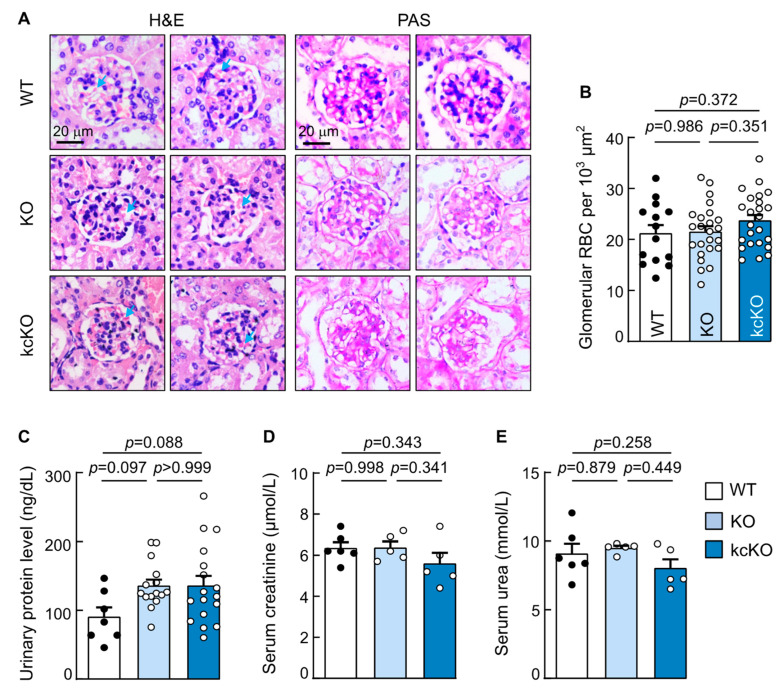
Analysis of renal histology and function in corin kcKO mice. (**A**) Kidney sections from WT, corin KO, and kcKO mice were stained with H&E (two left columns) or PAS (two right columns) and examined under a light microscope. Blue arrows indicate red blood cells (RBCs). Scale bars: 20 μm. Data are representative of at least three experiments. (**B**) Glomerular RBCs were counted in kidney sections from WT (black dots), corin KO, and kcKO (white dots) mice. *n* = 14–24 per group. (**C**–**E**) Levels of urinary proteins (**C**) and serum creatinine (**D**) and urea (**E**) in WT (black dots), corin KO, and kcKO (white dots) mice were measured by a Bradford assay (**C**) or an automated clinical chemistry analyzer (**D**,**E**). *n* = 5–17 per group. Data presented are mean ± SEM. *P*-values were analyzed by one-way ANOVA (**B**–**D**) or Kruskal–Wallis test (**E**).

**Figure 4 ijms-23-11251-f004:**
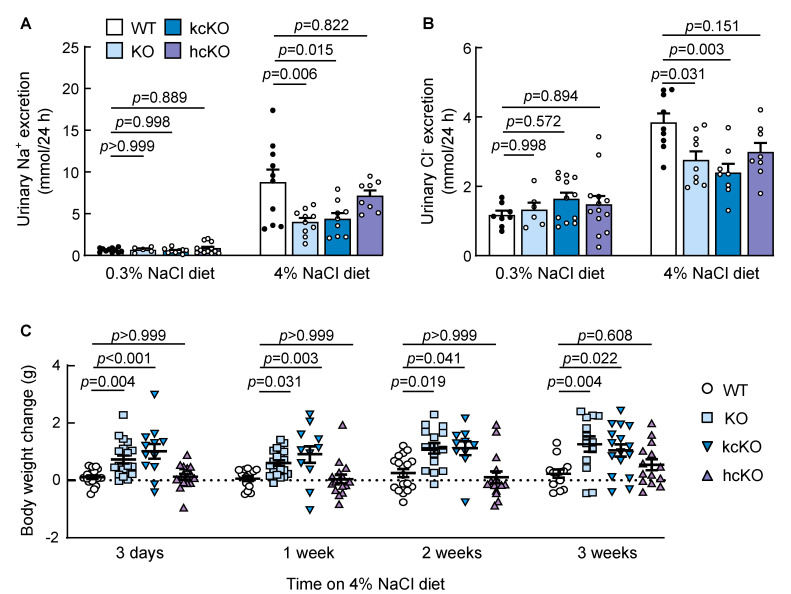
Urinary Na^+^ and Cl^−^ excretions and body weight change in corin kcKO mice. (**A**,**B**) Urinary Na^+^ (**A**) and Cl^−^ (**B**) excretion in WT (black dots), corin KO, kcKO, and hcKO (white dots) mice on 0.3% and 4% NaCl diet were measured. *n* = 5–13 per group. (**C**) Body weight changes in WT, corin KO, kcKO, and hcKO mice when 0.3% NaCl diet was switched to 4% NaCl diet. *n* = 10–21 per group. Data are mean ± SEM. *P*-values were analyzed by one-way ANOVA (**A**) or Kruskal–Wallis test (**B**,**C**).

**Figure 5 ijms-23-11251-f005:**
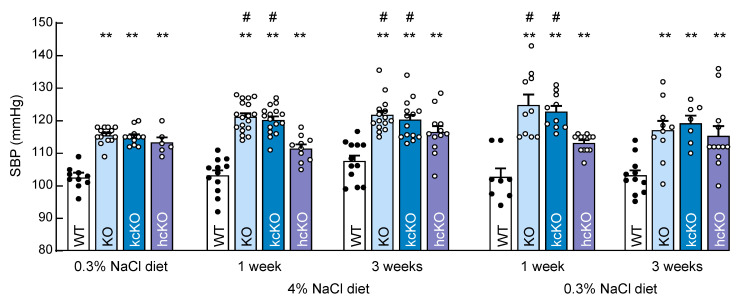
Salt-exacerbated hypertension in corin KO and kcKO mice. Systolic blood pressure (SBP) was measured in male WT (black dots), corin KO, kcKO, and hcKO (white dots) mice fed a 0.3% NaCl diet, then a 4% NaCl diet for one and three weeks, then back to a 0.3% NaCl diet for one and three weeks. ** *p* < 0.01 vs. WT of the same group. ^#^ *p* < 0.05 vs. the same genotype on initial 0.3% NaCl diet. *n* = 6–18 per group. Data shown are mean ± SEM. *P*-values were analyzed by one-way ANOVA.

**Figure 6 ijms-23-11251-f006:**
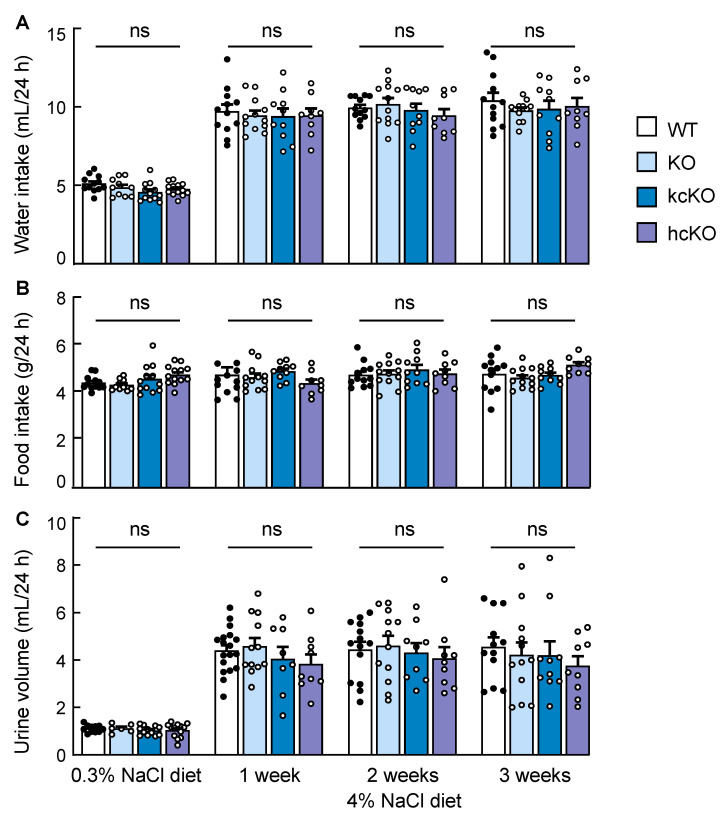
Water and food intakes and urine volume in corin KO, kcKO, and hcKO mice. (**A**–**C**) Water (**A**) and food (**B**) intakes and urine volume (**C**) were measured in male WT (black dots), corin KO, kcKO, and hcKO (white dots) mice on 0.3% and 4% NaCl diets for one to three weeks. *n* = 8–18 per group. Data shown are mean ± SEM. *p*-values were analyzed by one-way ANOVA (**A**) or Kruskal–Wallis test (**B**,**C**). ns, not significant.

**Figure 7 ijms-23-11251-f007:**
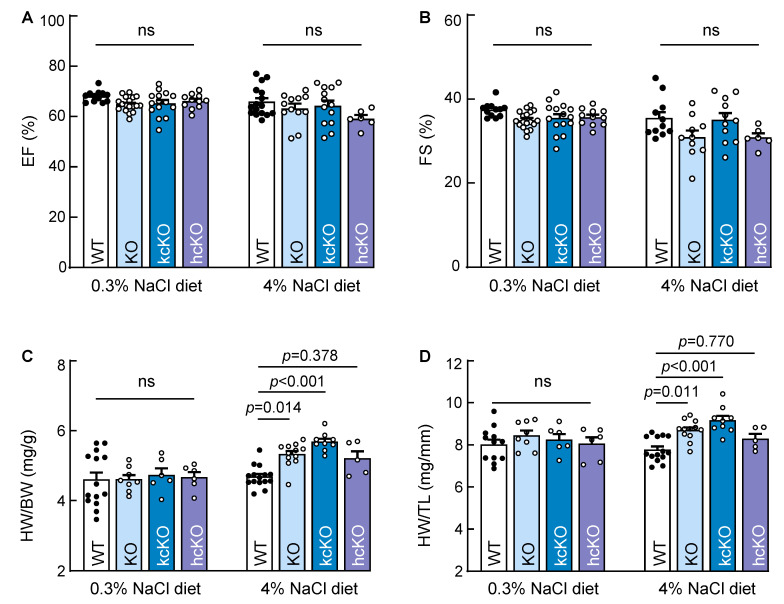
Cardiac hypertrophy in corin KO and kcKO mice on 4% NaCl diet. (**A**,**B**) Cardiac function, indicated by ejection fraction (EF) (**A**) and fraction shortening (FS) (**B**), was measured by echocardiography in male WT (black dots), corin KO, kcKO, hcKO (white dots) mice on 0.3% NaCl diet and 4% NaCl diet for three weeks. *n* = 6–18 per group. (**C**,**D**) Cardiac hypertrophy was assessed by ratios of heart weight (HW) to body weight (BW) (**C**) or tibia length (TL) (**D**) in male WT (black dots), corin KO, kcKO, hcKO (white dots) mice on 0.3% NaCl diet and 4% NaCl diet for three weeks. *n* = 5–15 per group. Data shown are mean ± SEM. *p* Values were analyzed by one-way ANOVA. ns, not significant.

## Data Availability

No large sets of sequencing or proteomic data were generated. All data that support the findings of this study are presented in the manuscript and the Appendix A of this article.

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
