# Peer review of "Renal Corin Is Essential for Normal Blood Pressure and Sodium Homeostasis"

_ijms, 2022, doi:10.3390/ijms231911251_

Round 1
Reviewer 1 Report
Despite the enormous amount of presented data, the current study is not correctly designed, and the manuscript is superficially written, omitting many crucial details. As a result, the conclusions and the manuscript title are overrated and mislead readers.
A major critical limitation of the current study that cannot be resolved by re-writing or incorporating additional experiments are:
- Lack of appropriate animal group control. The implication of WT C57BL/6J mice as a control group for Cre- conditional knockout groups of mice is unacceptable.
- Critical conclusion-driven experiments were performed under non-physiological conditions. 4% sodium diet is more than 13-fold above regular sodium consumption in humans and mice.
- Corin renal conditional knockout mice are questionable. Thus, corin is expressed in the proximal tubes, while the Ggt1 promoter delivers only to the cortical tubular epithelium of the kidney.
In addition, many other limitations must be addressed and satisfied before publication:
- All WT-type related data are widely distributed, which is unexpected in unstressed animals.
- Figure 1 B and C: Why have other molecular forms of corin not been detected?
- Critical measurements of non-cleaved pro-ANP forms are not correct. Fig. 1 F and D and Figure 2S related statements are wrong since the Elisa assay used for this data generation detects both cleaved and un-cleaved forms of pro-ANP.
- Figure 2B – unclear why cGMP circulating levels are so low in KO mice.
- BUN, creatinine, and serum urea data must be demonstrated for a 4% NaCl diet.
- Urine-specific gravity data must be provided for all groups and diets.
- What is the source of mouse diets?
Author Response
Please see the attached point-by-point response.

Reviewer 2 Report
The paper presents an appealing study showing that renal corin-mediated ANP activation is an important mechanism in enhancing natriuresis upon high salt intakes and that this function cannot be compensated by the cardiac corin and ANP-mediated endocrine mechanism in mice.
The paper is well written and the results of relevance both for research and practice.
Important strengths include the detailed physiological approach.
The implications for human health could be elaborated in more depth.
Do findings transfer to abnormal renal functioning?
How does the pattern of results help to better understand the link between renal and cardio-vascular diseases?
Which recommendations do the authors have regarding nutrition including salt; is this unique or specific for certain health profiles?
Author Response

(The authors gave the same response as above.)

Reviewer 3 Report
This paper by Zhou et al. assessed the role of renal corin in blood pressure and sodium homeostasis control in the mice. In renal corin knock-out mice high salt diet induced hypertension and cardiac hypertrophy by impairing Na+ and Cl- excretion, so providing evidence tha renal corin-mediated ANP activation is crucial for Na+ and Cl- homeostasis. Authors also conclude that this function cannot be compensated by the cardiac corin and ANP-mediated endocrine mechanism.
Specific Comments
1. The reason for hypertension in corin kcKO mice on 0.3% NaCl diet remains unclear, as stated by Authors themselves. This issue needs to be better argued.
2. Authors did not observe a reduction in urinary Na+ and Cl- excretion in corin hcKO mice on 4% NaCl diet. These results indicate that a renal corin and ANP-mediated autocrine or paracrine mechanism is crucial for promoting natriuresis upon high salt intakes and that this function cannot be surrogated by cardiac corin in mice. This evidence highlights a dysfunction in ANP metabolism that is also common to BNP metabolism, as previously demonstrated in some pathophysiological states in rat models (Holditch SJ et al., Kidney Int 2017; Tonne JM et al. Aging (Albany) 2014; Cataliotti A. et al., Circulation 2011) and human studies on blood pressure homeostasis (P. Belluardo et al., Am J Physiol 2006; F. Macheret et al, J Am Coll Cardiol 2012). These references should be brought to the fore to emphaasize the complex derangement of natriuretic peptides (ANP and BNP) metabolism underlying dysfunction in Na+ homeostasis and organ damage.
3. Line 233: hypertensin is misspelled and should run hypertension.
4. Fig.3, panel A: histological features should be highlighted by arrows.
Author Response

(The authors gave the same response as above.)

Round 2
Reviewer 1 Report
Dear Dr. Wu,
I agree with your arguments related to the Corin conditional knockout mice. Thank you.
Still, several critical issues remain and must be addressed or, at the very least, acknowledged as the critical limitations (in a separate Study Limitations sub-section) before manuscript reconsideration.
· Thank you for clarifying that Cre-conditional mice were crossed into the C57BL/6J background; please specify for how many generations. However, WT C57BL/6J mice are still not the correct experimental control for the current study.
· I understand that in animal experiments, 4-8% sodium diet is commonly used. However, it does not eliminate the point that this diet provides a sodium level that is around 10 folds higher than the average normal sodium consumption level in humans and mice and physiologically irrelevant. Please address this in the Abstract (line 19) and the Study Limitation section.
· Figure 1, B, C: It would be expected to see not-glycosylated or less glycosylated Corin zymogen forms of cardiac and renal Corin. As far as I can see, Figure 3a (Chen S. et al., Nat. Med. 2015, 21) demonstrates around 170 kDa forms of Corin in Wt and KO mice. Interesting that Abcam believes that ab255812 anti-corin antibody is unsuitable for WB application.
· Thank you for acknowledging the limitations of the used Elisa assay, which cannot detect pro-ANP processing. Please explain in the manuscript why Corin KO is expected to modulate the ANP-cGMP pathway in the tissue homogenates if Corin cleaves pro-ANP extracellularly upon secretion. Besides, why do you not consider that BNP (which is activated intracellularly by furin) may be responsible for cGMP level alterations?
· The cGMP data in heart tissue (Figure 1g) are not definitive. Therefore, they cannot (even collectively) support the conclusion that disruption of renal but not cardiac Corin gene impaired pro-ANP processing and cGMP generation in a tissue-specific manner (lines 85-89). Please provide definitive data or correct your conclusions (in Results, Discussion and Abstract).
· Thank you for providing the data on serum creatinine and urea from the 4% sodium diet-fed mice (Fig S7A, B). Please, move these data to Figure 3, to make it easy for readers to compare outcomes of 0.3% and 4% sodium diets; and explain in the related part of the manuscript why serum creatinine levels were almost twice lower in the mouse groups (WT and KOs) fed with 4% sodium diet vs. mice on 0.3% sodium diet.
· Differences in salt-exacerbated hypertension between cardiac KO and renal KO groups may be attributed to the different impact of a 4% sodium diet on stimulation of AngII production in these two mouse lines. Please present the related data to exclude the such possibility.
Reviewer 3 Report
No further concern.
Author Response
We thank the reviewer for helping us to improve our manuscript.
Round 3
Reviewer 1 Report
Dear Dr. Wu,
Thank you for addressing many of my questions and concerns and incorporating new sets of experimental data. The manuscript is now significantly scientifically improved.
Still, the critical limitation related to the experimental control remains unaddressed. Please acknowledge that the study did not use the Cre mouse as a control in the study limitations paragraph.
